# Asymmetric Reactions of *N*-Phosphonyl/Phosphoryl Imines

**DOI:** 10.3390/molecules28083524

**Published:** 2023-04-17

**Authors:** Devalina Ray, Suman Majee, Ram Naresh Yadav, Bimal Krishna Banik

**Affiliations:** 1Amity Institute of Biotechnology, Amity University, Sector 125, Noida 201313, UP, India; 2Amity Institute of Click Chemistry Research and Studies, Amity University, Sector 125, Noida 201313, UP, India; 3Department of Chemistry, Faculty of Engineering and Technology, Veer Bahadur Singh Purvanchal University, Jaunpur 222003, UP, India; 4Department of Mathematics and Natural Sciences, College of Sciences and Human Studies, Prince Mohammad Bin Fahd University, Al Khobar 31952, Saudi Arabia

**Keywords:** asymmetric synthesis, *N*-phosphoryl/phosphonyl imines, stereoselective, chiral catalysis, ligands

## Abstract

The asymmetric reactions of imines continued to attract the attention of the scientific community for decades. However, the stereoselective reactions of *N*-phosphonyl/phosphoryl imines remained less explored as compared to other *N*-substituted imines. The chiral auxiliary-based asymmetric-induction strategy with *N*-phosphonyl imines could effectively generate enantio- and diastereomeric amine, *α*,*β*-diamine, and other products through various reactions. On the other hand, the asymmetric approach for the generation of chirality through the utilization of optically active ligands, along with metal catalysts, could be successfully implemented on *N*-phosphonyl/phosphoryl imines to access numerous synthetically challenging chiral amine scaffolds. The current review critically summarizes and reveals the literature precedence of more than a decade to highlight the major achievements existing to date that can display a clear picture of advancement as well drawbacks in this area.

## 1. Introduction

Chiral and achiral imines portray a unique class of privileged substrate for efficient access to chiral amine analogues possessing remarkable significance in medicinal, synthetic, and natural-product chemistry [1,2,3,4,5,6]. Synthesis and isolation of chiral amines with high enantio- and diastereoselectivities have extensive applications in drug development, total and semisynthetic approaches in the natural product, chiral catalyst, ligand development, etc. [7,8,9,10]. In this regard, chiral amines can be produced by two independent approaches; first, chirality induced by an auxiliary group attached to imine and second, chiral catalysis in achiral imines [11,12,13,14,15,16,17]. Both of these approaches have been widely explored and adapted for the successful generation of stereoselective amines. The chirality-induced strategy required the introduction of a chiral auxiliary in the imine substrate, which has been well documented with *N*-sulfinyl imine and other related analogues [18,19,20,21,22,23,24,25]. However, the ease of protection and deprotection strategies involved with imine auxiliary is extremely important for the successful implementation of the proposed approach. Although being electron withdrawing, *N*-sulfinyl attached to imine provides an electrophilic centre as well as a configurationally defined stereogenic centre for predominant stereoselectivity and the use of *N*-sulfinyl imine is not easy [26]. The challenges associated with the deprotection of these amines, along with the instability under an oxidative environment, limit their application practically. All the efforts towards the resolution of the drawbacks end up compromising the stereoselectivity, as well as the retrieval, of chiral auxiliary, added to their intolerance for Lewis acids such as TiCl_4_, etc. [27]. In contrast, *N*-phosphonyl imine, an electrophile pursuing better stability than *N*-sulfinyl imine, leads to chiral amines with an excellent yield and higher stereoselectivity [28,29]. The electrophilicity of the imine centre can be curbed by incorporating electron deficient or electron-rich groups on the nitrogen atom. In this regard, G. Li and coworkers did pioneering work and pursued major goals with various chiral phosphonyl auxiliaries in numerous asymmetric transformations for the generation of chiral *N*-phosphonyl amines [30,31,32,33]. He further introduced the group-assisted purification (GAP) technology by washing with an appropriate solvent, for simple and time-efficient purification to isolate chiral *N*-phosphonyl amines with an improved yield and stereoselectivities [34]. The majority of protocols involving chiral *N*-phosphonyl imines reported by G. Li have been epitomized in this review.

The second strategy involves the active participation of either chiral organocatalysts or metal catalysts combined with chiral ligands for asymmetric induction in achiral *N*-phosphonyl/phosphoryl imine [35,36]. This approach led to the development of significant protocols for the stereoselective synthesis of amines, diamines, and other analogues [37,38]. In this area, Hoveyda et al. have contributed significantly through the development of chiral organocatalysts for efficient imine transformation [39,40,41]. The chiral *N*-phosphonyl/phosphoryl amines generated from corresponding imines can be easily deprotected to structurally-significant chiral amines, frequently existing in a medicinal and natural product framework [42].

The major collection of literature reports on this topic was accumulated till 2008 after which the critical analysis on the chiral synthesis of *N*-phosphonyl imine has not been documented categorically anytime in past [27]. Therefore, to fill the gap of knowledge, the present review intends to compile a major breakthrough by authenticating all the strategies developed from 2010 onwards to date. The content of the review has been subdivided further into two categories based on chirality induced either via *N*-phosphonyl imine auxiliary or chiral catalysis of achiral *N*-phosphonyl/phosphoryl imines for the sake of better understanding.

## 2. Asymmetric Reactions in Chiral *N*-Phosphonyl Imine Auxiliary

G. Li and coworkers hypothesized that (*S*)-BINOL-derivatized chiral *N*-phosphoryl imines underwent a facile 1,2-addition reaction with cyclic and linear diketones in the absence of a base to afford chiral amines (Figure 1) [43]. The starting imines, in turn, were synthesized from (*S*)-BINOL in a good yield through a series of reactions. These chiral phosphoryl imines also served as electron-deficient centres when reacted with diethyl malonate in the presence of potassium carbonate. Moderate to quantitative yield of products with high diastereoselectivities has been observed. The absolute configuration of the product was assigned in an indirect way by converting the product to an authentic compound.

The asymmetric addition of lithium glycine enolate to diphenyl diamine substituted *N*-phosphonyl imine furnished biologically-significant chiral *α*,*β*-diamino esters in good yields and high diastereoselectivities (Figure 2) [44]. The chiral *N*-phosphonyl imine auxiliary was accessed through convenient reaction in good yields. The absolute configuration of the deprotected *α*,*β*-diamino ester was determined by converting and matching with the literature-reported compound. A cyclic six-membered transition state was anticipated where an exceptionally *Si*-face approach fixes the C-3 configuration as *R*.

The configuration at C-2 responsible for anti-diasteoreselectivity can be assessed from the enolate isomer. The *Z*-enolate of glycine ester approaching the *Si*-face of chiral phosphonimines ends up in (2*R*, 3*R*) diamine.

Chiral *N*-phosphonyl propargylamines were easily accessed through the addition of lithium aryl/alkyl acetylides to *N*-phosphonylimines. The asymmetric addition was generalized with 17 substrates to furnish chiral products with an excellent yield and diastereoselectivities (up to 99:1). The role of bases in generating the acetylide anion, along with the solvent, was vital in dictating the outcome of the reaction. *N*,*N*-isopropyl substitution in the chiral *N*-phosphonylimines appeared as the ideal protecting option among others, along with the involvement of lithium metal in controlling the stereoselectivity. The absolute stereochemistry of the product through induced asymmetry was determined after the deprotection of the phosphonyl group. The absolute configuration was estimated as *S* since the product displayed optical rotation opposite in sign to that of the literature reported *R*-analogue (Figure 3) [45].

The chiral *N*,*N*-diisopropyl-*N*-phosphonyl imines underwent an aza-Darzens reaction for the successful asymmetric access to substituted aziridine-2-carboxylic esters with high diastereoselectivities. The replacement of the primary benzyl group with the secondary isopropyl group in *N*-phosphoryl imines was crucial and superior to other imine auxiliaries in accelerating the asymmetric environment in the reaction. The ^31^P-NMR analysis of the resulting crude products revealed the formation of only one isomeric product in most of the cases. The developed protocol established a simple and efficient approach to *β*-hydroxy-*α*-amino acids as well as various other amino building blocks. A unique operational procedure was executed by the gradual addition of the cold solution of imine to the in situ-generated *β*-bromolithium enolate mixture at −78 °C under an anhydrous condition in the presence of 4 Å molecular sieves.

A plausible mechanistic interpretation was outlined based on the previous methodology where a six-membered chair form of the transition state was anticipated. It was proposed that the Lewis acidic property of the lithium cation anchors both the imine electrophile and enolate nucleophile together prior to the aza-Darzens reaction. Consequently, the attack of enolate on imine’s carbon-centre-preferred *Re* face maintaining the stereochemistry. The stereo control was introduced indirectly through both the nitrogens of the phosphoryl counterpart connecting the (2*S*, 3*S*) vicinal centres on the cyclohexane ring (Figure 4) [46].

G. Li and coworkers escalated the asymmetric addition of carbamoyl anion to the chiral *N*-phosphonyl imines at low temperatures using LiHMDS for the synthesis of phenyl glycine amide derivatives (Figure 5) [47]. The application of the group-assisted purification (GAP) technique eliminated the requirement of column chromatography for the purification of the products. The chiral *N*-phosphonyl amides were generated in good yields (71–99%) and moderate to high diastereoselectivities (*dr* 84:16–95:5). The GAP ether washing lead to the improved yield of the diastereoenriched products. The absolute configuration of the product was fixed from X-ray crystallographic analysis, indicating the generation of (*S*) stereocenter. Following the lead, a six-membered transition state was hypothesized where the Lewis acidic lithium-ion coordinates to the phosphonyl oxygen, carbamoyl oxygens, and carbon simultaneously to form a tricyclic fused chair intermediate. The predominant 1,3-diaxial steric interaction of the isopropyl substituent existing below the ring with the axial R group forces the imine to get oriented in a manner that the R group stays in an equatorial position to minimize the steric hindrance. This in turn proceeds with the carbamoyl carbon attack from the *Re* face-generating (*S*) stereoisomer of the product.

An asymmetric [3 + 2] cycloaddition of chiral phosphonyl imine with methyl isocyanoacetate for easy access to 2-phosphonyl imidazoline with switchable stereoselectivity using base or silver salt was portrayed by G. Li and coworkers [48]. The cesium carbonate-mediated cyclization furnished stereoselective (4*R*, 5*S*) product with *dr* > 99:1, whereas AgF-catalyzed cycloaddition resulted in a change of the stereoselectivity leading to the (4*S*, 5*R*) product.

The controlled experiments led to the plausible mechanistic interpretations outlined in the scheme. Two divergent transition states were predicted with the base and silver salt undergoing [3 + 2] cycloaddition to afford diastereoselective products (Figure 6). The stereoselectivity of the base-catalyzed reaction was regulated by the attack of the nucleophilic isocyanate anion on the favourable *Si* face of the chiral phosphonyl imine to produce a diastereoselective intermediate that furnished diastereopure product after nucleophilic intramolecular annulation. An opposite stereoselectivity was reflected in the Ag-catalyzed reaction where the C–C bond formation is described through the formation of a six-membered chair conformation involving the simultaneous participation of imine nitrogen, carbon from isocyanide, and silver coordination stabilized by oxygen to phosphonyl. The deteriorating diasereoselectivity for the electron-deficient substrate is attributed to the reduced transition-state stability.

In continuation with the previous method Aza-Morita–Baylis–Hillman reaction was performed with chiral *N*-phosphonyl imines and acrylonitrile by G. Li and coworkers. The products obtained in good to excellent yield and high diastereoselectivity were purified with a GAP (group-assisted purification)-applying cosolvent, hexane: ethyl acetate (10:1) (Figure 7) [49]. Based on previous reports, and the depicted conformation of the product, the initial step was hypothesized as the Michael addition to the acrylonitrile forming the zwitterionic intermediate followed by the addition of chiral *N*-phosphonyl imines. The stereoselectivity was justified by a nucleophilic attack from the Re face of the *N*-phosphonyl imines to provide a second zwitterionic adduct. The consequent transfer of protons, followed by the elimination of Bu_3_P, afforded a diastereorich product. The reuse of eliminated Bu_3_P initiates the catalytic cycle and continues till completion.

The asymmetric Mannich reaction of azlactone with *N*-phosphonyl imine was demonstrated by G. Li and coworkers at an ambient temperature under mild conditions in the absence of any catalyst, base, or additive (Figure 8) [50]. The products were achieved with excellent yield and high diastereoselectivities. The *syn*-diamino acid with α-quarternary carbon obtained as a product was purified by employing a group-assisted purification (GAP) technique through washing with cosolvent. The diastereopure products were transformed further to obtain valuable fragments of certain bioactive peptides and medicinally important pharmacophores. The previous reports reveal the reaction to proceed with an aza-ene reaction where the oxygen linked to the phosphorus of the *N*-phosphonyl imine binds with the enol form of azlactone via hydrogen-bond interaction to generate the intermediate. The subsequent aza-ene reaction forms the diastereoselective products where the chiral auxiliary plays a pivotal role in controlling the stereochemistry.

G. Li and coworkers established an effective protocol for the synthesis of chiral α-amino phosphonates in high yields and diastereoselectivities (up to 99:1) via hydrophosphorylation of chiral *N*-phosphonyl imines with the phosphites in the presence of lithium metal (Figure 9) [51]. The bases implemented for the generation of an active nucleophile are critical for chiral induction. The chiral α-amino phosphonates were converted into the known Cbz-N derivatives whose optical rotations were matched to obtain the absolute configuration of the product. The product configuration was determined to be *R,* through the attack of nucleophilic lithium phosphite to the *N*-phosphonyl imines from *Re*-face. Diverse *N*-phosphonyl auxiliary was examined to evaluate the effects of *N*,*N*′-R groups. Among isopropyl, isobutyl, neopentyl, cyclopentyl, benzyl, and CH_2_-1-naphthyl groups, the exclusive diastereoselectivities and high yield of the products were obtained with isopropyl-substituted chiral *N*-phosphonyl imines.

An achiral umpolung reaction of *N*-phosphonyl imines and 2-lithio-1,3-dithianes furnished a high yield of *α*-amino-1,3-dithiane in excellent diastereoselectivities of more than 99:1. The rate of addition of the substrates in the reaction controls the diastereoselectivity. The chiral *α*-amino-1,3-dithiane substituted *N*-phosphonyl imines were purified through GAP chemistry by washing out the crude product with a cosolvent where the need for traditional chromatography can be circumvented. The stereochemistry of the umpolung product was concluded by HBr/MeOH mediated cleavage of the *N*-phosphonyl group and subsequent tosylation of the free amine to furnish a known compound having an *R*-configuration.

A chair form of a six-membered cyclic transition state was hypothesized to justify the stereoselectivity observed and supported by previous reports. However, the specific stereoselectivity of the chiral products to the *R* configuration contrasted with the similar category of reported addition reactions in enolates. Therefore, this can be considered as the only instance where a reversed control for the formation of chiral *N*-phosphonyl amines occurred using (*R*,*R*)-1,2-cyclohexanediamine auxiliary. In comparison to the prior asymmetric transformation, the present reactions seem to be unique in terms of the mechanistic approach where lithium is coordinated with the oxygen of *N*-phosphonyl imines instead of the coordination of cationic lithium to nitrogen of the same imine (Figure 10) [52].

The design, synthesis, and application of novel *N*-phosphinyl imines and *N*-phosphinimide in an asymmetric Aza-Henry reaction were exemplified by G. Li and coworkers (Figure 11) [53]. The asymmetric reaction of the chiral imines with nitromethane furnished the products with high yields (92–99%) and excellent diatereoselectivity (>99%). The substrate scope for the formation of the various substituted aldehyde and ketone-derived chiral imine as electrophiles appeared to be vast enough. The chiral *N*-phosphinyl imines were stable for a longer time at room temperature and inert atmosphere. The products could be isolated through GAP technology by the washing of the crude mixture with ethyl acetate and hexane. The product configuration was determined by deprotection with the removal of the chiral *N*-phosphinyl group from the product and subsequent conversion of *β*-nitroamine to the known *N*-Boc derivative. A plausible mechanism was anticipated based on the product configuration where the metallic lithium binds with the imine nitrogen and fixes the two reactant counterparts forming a chair-like transition state. Thereafter, the attack of nucleophilic anionic nitromethane to chiral *N*-phosphinyl imine occurs from *Re*-face generating the *S*-stereo centre.

The chiral *N*-phosphonyl imine primarily developed and reported by G. Li and coworkers served as excellent electrophiles to get attacked by cyanide ion from diethylaluminium cyanide, a nonvolatile and inexpensive source which makes it a greener alternative (Figure 12) [54]. The Strecker reaction of *N*-phosphonyl imine and diethylaluminium cyanide along with isopropanol as a coadditive produced α-aminonitriles in moderate to high yields (up to 98%) and excellent diastereoselectivities (up to 99%). The synthesis and purification of products using GAP chemistry eliminated the requirement of traditional separation techniques and simplified the process by washing of crude products with hexane.

A logical mechanistic interpretation was framed based on the absolute configuration of deprotected *α*-aminonitrile with *S*-stereocentre. Initially, Et (*^i^*PrO)AlCN formed from Et_2_AlCN and *^i^*PrOH approaches the *Re*-face of chiral *N*,*N*′-isopropyl-*N*-phosphonyl imine. The chair-form transition state was devoid of any interaction of imine nitrogen with Et_2_AlCN, as the use of excess Et_2_AlCN was restricted in the reaction. Two minor steric aspects, i.e., hydrogen and the lone pair of nitrogen exist in the cyanide incorporation pathway. However, the chiral synthetic version assures the *S*-chirality among other possibilities.

An asymmetric borylation of chiral *N*-phosphinylimines was executed through GAP chemistry for efficient access to Velcade by G. Li and coworkers (Figure 13) [55]. Among the seven steps in the synthesis of this anticancer drug, the generation of chiral imine and its borylation followed the GAP chemistry through an easy workup and purification steps avoiding column chromatography. The optically pure and diastereorich product could be obtained by simply washing with hexane, whereas the chiral phosphonyl auxiliary can be simply deprotected and recovered quantitatively. The X-ray diffraction analysis predicted the absolute configuration of the borylated product.

An asymmetric 1,2-addition of the allyl magnesium reagent to the chiral *α*,*β*-phosphonyl imines was emphasized by J. Han & Y. Pan to provide direct access to chiral α-alkenyl homoallyl amines (Figure 14) [56]. Regioselective formation of 1,2-adduct was observed with all the substrates in high yields and stereoselectivities. The synthesis of chiral *α*,*β*-unsaturated imines were initially optimized through the condensation of phosphoramide and cinnamaldehyde. Thereafter, the 1,2-addition of the Grignard reagent to *α*,*β*-phosphonyl imines afforded diastereorich phosphonyl protected α-alkenyl homoallyl amines. The chiral phosphonyl auxiliary was removed efficiently to produce amine hydrochloride salts. The absolute configuration was detected as an *S*-isomer by matching with the known sample.

In continuation with the previous protocol, G. Li and co-workers further extended the concept through the asymmetric synthesis of chiral phosphonyl amines by 1,2-addition of various Grignard reagents to aliphatic *N*-phosphonyl hemiaminal (Figure 15) [57]. The alkylated homoallyl amines could be generated in high yield and diastereoselectivity through a three-component reaction of *N*-phosphonyl amide, aliphatic aldehydes and allyl magnesium halide performed in a single pot. The use of alkyl or aryl Grignard reagents other than allyl provides the desired *N*-phosphonyl amide in good yields where the diastereoselectivity was compromised. The variation in mechanism for both approaches was responsible for a different degree of stereoselectivities. It was depicted that the in situ hemiaminal formed which was not isolated further undergoes hydrogen elimination from nitrogen and isopropyl functionalities under a basic condition. The resulting *N*-phosphonyl imine was allylated by the attack from a less hindered side via the formation of a six-membered transition state followed by a double bond displacement to afford (*S*)-phosphonyl amide.

In contrast, the *S*-configuration with other alkyl and aryl reagents ensues out of their attack into the imine from the *Si* face through a metal–oxygen coordination leading to a different six-membered transition state. In this case, the position of the prochiral centre was significantly distant from the auxiliary to face the predominant effect of the steric group that dictates the selectivity. Hence, low diastereoselectivity of products was observed in this case.

## 3. Asymmetric Reactions in Achiral Phosphonyl Imine Using Chiral Catalyst/Ligand

The asymmetric reaction of achiral phosphonyl imines under organocatalytic or metal-catalyzed conditions successfully afforded chiral products with considerable stereoselectivities. In this direction, novel reaction protocols have been introduced by a few groups in the past decade. Hoveyda et al. introduced an exceptionally novel and unique concept of small molecule chiral organocatalysts for the highly enantioselective formation of amines and alcohols from achiral imine and carbonyl substrates (Figure 16) [58]. The addition of unsaturated organoboron reagents to achiral phosphonyl imines generated enantiorich pure-alkenyl amines which can serve as an intermediate toward easy access to bioactive analogues. This operationally simple and facile protocol could afford enantiopure products in an 87% yield and 97:3 enantioselectivity. The substituted chiral aminophenol, which can be modulated and manipulated structurally, was introduced as an efficient catalyst. The reaction of achiral (pinacolato) allyl boron can be converted to chiral allyl boron by coordination with the nitrogen of the chiral amino alcohol. However, at the same time, the boron atom connected to the electron-rich nitrogen of amine needs to have enough Lewis acidity to coordinate with the substrate. An alternative way out is the intramolecular hydrogen coordinating the amide carbonyl of the catalyst counterpart and boron attached to nitrogen, which enhances the Lewis acidity and substrate-binding affinity.

The same research group once again introduced the highly enantioselective formation of homoallylic amines via the metal-catalysed addition of stabilized and accessible (pinacolato) allyl boron to various aryl/heteroaryl and alkyl/alkenyl substituted *N*-phosphinoylimines (Figure 17) [59]. The Cu–NHC ligand complex synthesized from symmetric imidazolinium salts is crucial for the asymmetric transformation with allylic substrates to afford a quantitative yield of products in 98.5:1.5 *er*. The protocol is applicable for gram-scale synthesis, maintaining the yield and *er* proportionately. The mechanistic exemplification is based on a few critical observations where the quantity of the MeOH and Cu–NHC complex was varied and analyzed. It was predicted that the reaction of the NHC–Cu allyl complex is faster with aldimines than incidental protonation of the allylmetal intermediate’s carbon–copper bond. The stereochemical predominance was the outcome of preferred conformation in the transition states. The Lewis basic oxygen of *N*-phosphinoylimine coordinating with Cu appeared as the critical contact point between the substrate and the Cu-catalyst, whereas the nitrogen being relatively less basic might have a weak linkage with copper. The spatial orientation of the aryl group in aldimine while coordinating with Cu via the oxygen atom of the phosphinoyl counterpart controls the enantioselectivity of the products. The steric crowd arising out of the repulsion between the methyl substituent of NHC and the phenyl group in *N*-phosphinoylimine is far more predominant than that with the aryl substitution in aldimine. The difference is due to the labile binding of nitrogen of aldimine to the Cu-catalyst rather than the relatively stronger coordination of the oxygen atom from the same molecule with the metal.

Catalytic synthesis of chiral *α*-branched amines proficiently and stereoselectively contributes to the generation of numerous vital pharmaceutically and medicinally active compounds. In this context, homopropargyl amine analogues are significantly applicable in the semi-synthesis and total synthesis of many natural products. Consequently, the chiral auxiliary approach has been espoused widely for this purpose to obtain the product in great diastereoselectivity, whereas the analogous catalytic procedures are very limited. In this direction, the same group of A. H. Hoveyda has represented a novel and robust Cu-chiral *N*-heterocyclic–carbine complex-catalysed protocol for the generation of homopropargyl amines with excellent enantoselectivities. A diverse array of aryl/heteroaryl, alkenyl and alkyl-functionalyzed *N*-phosphinoyl imines underwent facile reaction with allenyl boron to afford chiral homopropargyl amide up to 98% yield and 98:2 *er.* in less than 7 h (Figure 18) [60].

The mechanistic approach for the Cu–NHC catalysed asymmetric transformation with allenyl substrates involves the formation of the Cu-allenyl intermediate in its initial step. The complex intermediate instigates from the Cu-alkoxides either through the metathesis of the σ-bond in allenyl boron or by the ligand exchange of metal alkoxide with allenyl boronate. The energetically favourable Cu-allene, as compared to its more nucleophilic Cu-propargyl analogue, is preferable for furnishing allenyl amide. The favoured geometry for the reaction portrays the coordination with the substrate from the most accessible face of the complex which encounters exceptional enantioselectivity.

A facile organocatalytic enantio- and diastereorich transformation of *N*-phosphoryl imine to *β*-nitro ethylphosphoramidates was established by Z. Miao and coworkers (Figure 19) [61]. The nitro-Mannich reaction of *α*-alkyl/aryl/heteroaryl/alkenyl nitroacetates with *N*-phosphoryl imine catalysed by thiourea derived from cinchona alkaloid was performed in toluene at low temperature. The established method for the asymmetric reaction was compatible with most of the substituted *N*-phosphoryl imine rendering products with adjacent tertiary and quarternary chiral centres in an excellent yield with excellent enantio- (99% *ee*) and diastereoselectivities (99:1 antiselectivity). The substrate scope was thoroughly identified with diverse electron-withdrawing or electron-donating substitution on the aryl functionality in *N*-phosphoryl imine. 

The mechanistic interpretation was predicted and supported by monitoring the shift in the peak of ^31^P NMR from the *N*-phosphoryl imine to *α*-substituted nitroacetates. The initial ^31^P NMR peak for *N*-phosphoryl imine in toluene appeared at 7.68 ppm. The addition of the thiourea catalyst followed by 2-nitropropanoate further led to a change in the peak position to 8.26 ppm assigned for the intermediate where thiourea –NH is hydrogen bonded to the oxygen atom of the phosphoryl imine, along with an emerging new peak at 5.67 ppm. The new peak accounts for the appearance of a new intermediate where the two substrates along with the thiourea catalyst are connected to each other preferably through hydrogen bond interaction via the *Si*-face attack of 2-nitropropanoate to the thiourea-activated *N*-phosphoryl imine. With the elapsing time, the peak at 5.67 ppm intensified with the gradual disappearance of the starting material, as indicated by the vanishing peak of ^31^P NMR at 7.68 ppm. The reaction was completed in 17 h, as indicated by the ^31^P NMR spectrum.

Y.-J. Xu and L. Dong have recognized an iridium(III)-catalyzed sequential C-H activation followed by [3 + 2] cycloaddition for direct access to spirocyclic phosphoramides (Figure 20) [62]. It was realized that [Cp*IrCl_2_]_2_ executed exceptional performance in terms of efficiency and selectivity in the chiral cascade reaction. The iridium-catalyzed asymmetric reaction of cyclic *N*-phosphoryl ketoimines and alkynes in the presence of the Cu salt additive furnished spirocyclic phosphoramide in complex heterocyclic systems.

A diverse array of sterically and electronically dissimilar *N*-phosphoryl ketoimines were explored as the coupling counterpart to scrutinize the generality of the reaction condition providing stereoselective products in high yield and diastereoselectivity (*dr* > 99:1)

The establishment of chiral pyridoxal-aldehyde catalysts in the asymmetric Mannich reaction by Zhao and coworkers discloses an avenue of novel synthetic strategies through chiral organocatalysis. Thereafter, Chen and Enders demonstrated that the novel chiral pyridoxal-aldehyde catalyst delivers an efficient pathway for the stereocontrolled addition of amines to the *N*-phosphinyl imines for the generation of structurally and conformationally important diamino acid esters having extensive biological and pharmaceutical significance. The crucial steps in the mechanistic approach have been depicted and outlined in Figure 21 [63]. The role of the catalyst initiated the formation of aldimine intermediate through the condensation of pyridoxal-aldehyde and *tert*-butyl glycine. Further deprotonation of the acidic C-H bond led to the generation of stable a carbanion which endured a facile 1,2-addition to the electrophilic centre of *N*-diphenylphosphinyl imine, forming a stable adduct. The hydrolysis of these adducts formed the product, efficiently liberating the pyridoxal-aldehyde catalyst in its active form to promote the catalytic cycle again. It was assumed that the pendent group in the catalyst plays an extremely vital role in linking the carbanion to the electrophilic centre of the imine.

The asymmetric reactions of *N*-phosphonyl imines were further explored by M. Zhang and coworkers. In this context, a bifunctional chiral diamine system was explored for asymmetric Mannich reaction of glycine iminoesters (Figure 22) [64]. The metal catalysis pooled with organocatalysis can lead to the efficient construction of two vicinal stereocentres. An enormous collection of syn-diamino esters was obtained in very good yield and excellent diatereo- and enantioselectivities (*dr* > 99:1 and 99% *ee*) through Cu(I)/amidophosphine–urea complex-catalysed Mannich reaction of glycine iminoesters with *N*-phosphinoyl imines. The overall absolute and relative configuration of the product was assessed from the X-ray crystallography. A plausible mechanistic pathway was framed via a hypothetical transition state where glycine iminoesters and *N*-phosphinoyl imines are simultaneously activated synergistically by the Cu–ligand complex. A tetracoordination is established between the Cu and the in situ generated enolate of glycine where the distance of the *N*-alkylidene group from the phenyl substituents of phosphorus is large enough to avoid steric congestion. Additionally, the P=O bond of *N*-phosphinoyl imines possesses dual activation through a hydrogen bond interaction with two –NH units of urea. As a result, the Cu coordinated iminoester approaches the electrophilic imine centre from the *Si*-face with a predominant formation of (*2S*, *3R*)-products.

A highly syn-selective Mannich reaction involving α-isocyanoacetates and *N*-phosphinoyl ketimines has been accentuated by Nakamura and coworkers using a Cu(II)-cinchona alkaloid-based catalyst [65]. The established protocol demonstrates direct and easy access to congested imidazoles containing vicinal trisubstituted and tetrasubstituted stereocentre in great yield with excellent diastereo- and enantioselectivity. The pseudoenantiomeric chiral combination of cinchona alkaloid and Cu(OTf)_2_ catalyst along with caesium carbonate as the base in THF successfully delivered the enantiorich *syn*-product. This stereodivergent synthetic strategy could provide stereorich *α*,*β*-diamino acid with sterically crowded adjacent quarternary stereocentres. The cinchona ligand combines with Cu and Cs_2_CO_3_ to form a complex with the hemiacetal of ketenegenerated from *α*-isocyanoacetates. Thereafter, the *N*-phosphinoyl ketimine gets linked to the complex and reacts with it in the coordination sphere of Cu resulting in a Mannich product through carbon–carbon bond formation. The cyclization of the Mannich product via imidazoline formation followed by ligand interchange between imidazoline and *α*-isocyanoacetates afforded a product with a newly generated (4*R*, 5*R*) stereogenic centre (Figure 23).

A similar stereodivergent Mannich-type reaction was investigated by Shibasaki and coworkers for a direct and efficient catalytic reaction of aryl/heteroaryl methyl *N*-phosphinoyl ketimines and *α*-methyl-*α*-isothiocyanato ester [66]. The switch in diastereoselectivity of the products can be achieved by the alteration of the metal catalyst. Sr catalyst/ligand furnished antiproducts in up to 97% *ee* and 4:96 *d.r.* (syn/anti), whereas the Mg catalyst/ligand gave syn products with up to 95% *ee* and 93:7 *d.r.* (syn/anti) (Figure 24a,b). The mechanistic insight for the stereodivergent product formation with the change in the catalyst was attributed to the difference in the aggregate formation of the two catalysts displayed through circular dichroism (CD). Chiroptically distinct aggregates were supposed to be generated from each metal–ligand solution. Moreover, the dihedral angle of the naphthylene counterpart in the metal–ligand for Mg and Sr is different, which is responsible for stereodiscerning the asymmetric step, resulting in variation of the diastereoselectivities.

## 4. Conclusions

In conclusion, the asymmetric transformation of imines possesses extensive future scope for successful utilization in addressing various scientific challenges, witnessing the past developments and their application in this area. Among them, the chemistry of *N*-phosphinoyl/phosphoryl imines and the variations in their stereoselective reactions are of ultimate interest to the scientific community for various reasons: (i) the stereo-rich *N*-phosphoryl amine derivatives obtained from various asymmetric transformations of corresponding imines existing to date, are the prime scaffold of importance in the medicinal, pharmaceutical, total, and semisynthetic transformations of natural products; (ii) The asymmetric reaction of chiral auxiliary-based *N*-phosphoryl imines or chiral catalysis-based reactions in the achiral phosphoryl imines substrates are equally or more efficient and stereoselective to that of the other reported reactions in imines; (iii) as per the requirement, the installation, removal, and recovery of the chiral phosphoryl auxiliary in *N*-phosphinoyl/phosphoryl imines is quantitative and completely stereoselective, retaining the original configuration of the amine stereogenic centre formed during chiral induction; (iv) in general, the asymmetric reactions of chiral and achiral *N*-phosphinoyl/phosphoryl imines are significantly less explored to date than other chiral imine transformations. Although there are numerous advantages associated with the stereoselective reaction in *N*-phosphinoyl/phosphoryl imines, certain drawbacks need to be addressed in the near future to overcome the hurdles in their successful implementation to various existing and upcoming areas. In this context, the variations and screening of phosphoryl auxiliaries need to be more extensive to fine-tune the stereoselective transformations. Additionally, as limited asymmetric reactions of phosphoryl imines have been documented in more than the past 14 years, colossal effort and emphasis on the specific topic should be devoted in terms of research progression and literature compilation. The bourgeoning novel concepts on the asymmetric transformations in *N*-phosphinoyl/phosphoryl imines will surely foster and flourish the synthetic strategies of this unique structural entity in the near future.

## Data Availability

Not applicable.

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
