# Peer review of "Asymmetric Reactions of N-Phosphonyl/Phosphoryl Imines"

_molecules, 2023, doi:10.3390/molecules28083524_

Round 1

Reviewer 1 Report

In this review, the authors came up with two significant strategies for the asymmetric reactions of imines. Although the strategies are well explored, it is quite important to be aware of updated knowledge of this area for the betterment of the synthetic community which is why the authors tried their best to fill the knowledge gap by presenting the literature from the year 2010. The presentation is flawless and transparent. Almost tried to cover everything happened in the past literature including the plausible mechanistic proposals and studies mentioned in each and every reported publication. It is very easily understandable and highly potential concepts were included. Hence, it is recommended to publish in molecules as such.

Author Response

Reviewer 1:

  1. In this review, the authors came up with two significant strategies for the asymmetric reactions of imines. Although the strategies are well explored, it is quite important to be aware of updated knowledge of this area for the betterment of the synthetic community which is why the authors tried their best to fill the knowledge gap by presenting the literature from the year 2010. The presentation is flawless and transparent. Almost tried to cover everything happened in the past literature including the plausible mechanistic proposals and studies mentioned in each and every reported publication. It is very easily understandable and highly potential concepts were included. Hence, it is recommended to publish in molecules as such.

Response: Thanks to reviewer for the kind recommendation to publish in Molecules as such.

Reviewer 2 Report

The use of N-phosphonyl and phosphoryl imines in asymmetric synthesis does not appear to have been widely exploited and this review could make their potential more widely known. Recent work in this area is limited to relatively few groups and this is reflected in the fact that 30% of the references are to work by just one research group. The authors have covered the recent literature in a satisfactory manner with detailed and clear accompanying figures. Although it is claimed that the review critically summarizes major achievements, there are only some rather general comments concerning advantages over alternative procedures. 

The text of the submission would benefit greatly from being checked by proof-reader proficient in English. In particular, a more correct title of the paper would be "Asymmetric Reactions with N-Phosphonyl/ Phosphoryl Imines" or "Asymmetric Reactions using N-Phosphonyl/ Phosphoryl Imines". There are a large number of other linguistic errors in the text and, while most of these do not prevent an understanding of the paper, there are some which are either unjustified, such as "unprecedented" significance (page 1), or problematic/unusual such as "launching" of stereoselective amines (page 1), metal catalysts "clubbed" with chiral ligands (page 2), "coping up" with various scientific challenges (page 25). Also, there are several instances of named reagents or procedures where the name has not been properly capitalized, e.g. lewis instead of Lewis, mannich instead of Mannich, grignard instead of Grignard etc.

The chemical structures are well drawn but there are some inconsistencies in the labelling of the aryl groups. In some schemes, the positions of the substituents are denoted by o-, m-, or p- whereas in others numbers are used. also insufficient are has been given to correctly indicating the substituents. For example, in Scheme 1, Ar = 4-OMeC6H4 should be written 4-MeOC6H4; in Scheme 2, 4-CNC6H4 would imply isocyanophenyl whereas presumably it refers to cyanophenyl, i.e. 4-NCC6H4. Naphtha-4-yl should also be corrected to naphth-2-yl; in Scheme 5, FPh and BrPh are not proper notations and should be given as FC6H4 and BrC6H4; in Scheme 10, entry 8 should be 4-BnO-Ph.

Author Response

Reviewer 2:

  1. The use of N-phosphonyl and phosphoryl imines in asymmetric synthesis does not appear to have been widely exploited and this review could make their potential more widely known. Recent work in this area is limited to relatively few groups and this is reflected in the fact that 30% of the references are to work by just one research group. The authors have covered the recent literature in a satisfactory manner with detailed and clear accompanying figures. Although it is claimed that the review critically summarizes major achievements, there are only some rather general comments concerning advantages over alternative procedures. 

Response: As pointed by the reviewer very correctly that there are limited reports on the asymmetric reactions of N-phosphonyl and phosphoryl imines in literature till date and among them, most of the reports on the use of N-phosphonyl group in imines as chiral auxiliary was pursued by G. Li and coworkers. Moreover, the reviews and perspectives focussing on this area are almost negligible. The review has been aimed to compile and discuss on both the advantages and drawbacks of this asymmetric approach. It also highlights the future scope and improvements required in this field. Therefore, we believe that it critically discusses the advancements of N-phosphonyl and phosphoryl imines to some extent.

  1. The text of the submission would benefit greatly from being checked by proof-reader proficient in English. In particular, a more correct title of the paper would be "Asymmetric Reactions with N-Phosphonyl/ Phosphoryl Imines" or "Asymmetric Reactions using N-Phosphonyl/ Phosphoryl Imines". There are a large number of other linguistic errors in the text and, while most of these do not prevent an understanding of the paper, there are some which are either unjustified, such as "unprecedented" significance (page 1), or problematic/unusual such as "launching" of stereoselective amines (page 1), metal catalysts "clubbed" with chiral ligands (page 2), "coping up" with various scientific challenges (page 25). Also, there are several instances of named reagents or procedures where the name has not been properly capitalized, e.g. lewis instead of Lewis, mannich instead of Mannich, grignard instead of Grignard etc.

Response: As per reviewer comment, the possible changes in the language has been incorporated in the manuscript. The title has been altered to “Asymmetric Reactions of N-Phosphonyl/ Phosphoryl Imines” as suggested by reviewer. The term "unprecedented" significance has been changed to “remarkable” significance whereas "launching" of stereoselective amines has been replaced by “generation” of stereoselective amines. Furthermore, metal catalysts "clubbed" with chiral ligands (page 2) has been changed to metal catalysts "combined" with chiral ligands. The part of statement such as "coping up" with various scientific challenges (page 25) has been changed to "addressing" various scientific challenges. The mistakes in named reaction and reagents throughout the manuscript have cross checked and rectified.

  1. The chemical structures are well drawn but there are some inconsistencies in the labelling of the aryl groups. In some schemes, the positions of the substituents are denoted by o-, m-, or p- whereas in others numbers are used. also insufficient are has been given to correctly indicating the substituents. For example, in Scheme 1, Ar = 4-OMeC6H4 should be written 4-MeOC6H4; in Scheme 2, 4-CNC6H4 would imply isocyanophenyl whereas presumably it refers to cyanophenyl, i.e. 4-NCC6H4. Naphtha-4-yl should also be corrected to naphth-2-yl; in Scheme 5, FPh and BrPh are not proper notations and should be given as FC6H4 and BrC6H4; in Scheme 10, entry 8 should be 4-BnO-Ph.

Response: All the mistakes mentioned has been noted and corrected according to reviewer’s suggestion. The positions of the substituents have been denoted by numbers throughout the manuscript. The notation of substituents has been rectified.

Reviewer 3 Report

The review is well written and it can be important for readers on the field of phosphorus chemistry. There are some minor issues:

-Scheme 17: the scheme of the imidazolium salt (right up) is wrong. There is no H at the nitrogen (MesHN-ring)
-There are many proposed mechanisms, but some of them should be discussed in more detail. E.g, there are some studies, where the mechanism was investigated by DFT calculations (e.g. ref 24). These mechanistic investigations should be extended.

Author Response

Reviewer 3:

  1. The review is well written and it can be important for readers on the field of phosphorus chemistry. There are some minor issues:

-Scheme 17: the scheme of the imidazolium salt (right up) is wrong. There is no H at the nitrogen (MesHN-ring)
-There are many proposed mechanisms, but some of them should be discussed in more detail. E.g, there are some studies, where the mechanism was investigated by DFT calculations (e.g. ref 24). These mechanistic investigations should be extended.

Response: As per reviewer’s comment the imidazolium salt structure has been corrected. Most of the mechanistic pathway has been detailed schematically as well as in text. The mechanistic interpretation based on Circular Dichroism (CD) outcome and other investigations has been incorporated in the manuscript for Scheme 24 (DFT study is not given in the related paper) as per reviewer’s suggestion.

Reviewer 4 Report

This review summaries asymmetric reactions of N-phosphonyl/phosphoryl imines developed by Li, Hoveyda and few other groups in the past decade, including the chiral auxiliary based induction strategy and asymmetric catalysis, to effectively access to enantio and diastereomeric amines. Given importance of chiral amines in synthesis of medicinal, pharmaceutical, and natural products, I believe this review will appeal to a wide range of readers and recommend a publication in Molecules. However, there are many grammar and format errors to be corrected before publication although the manuscript appears to be well-organized. For examples but not limited to,

1) The use of the singular and plural appears to be rather casual thorough out the manuscript. Too many to be listed here.

2) “The challenges associated with deprotection of these imines……”? should be “…… deprotection of these amines.”

3)  “significant yield of……” is not common.

4) Many grammar errors like “Diverse N-phosphonyl auxiliary were examined…..” etc.

5) Please capitalize the first letter in “grignard reagent” thorough out the manuscript.

6) Please check the capitalization thorough oue the manuscript, e.g. “concept through Asymmetric synthesis”, “application of Group-assisted purification” …… etc.

7) “……stereoselectivity, hence owing to the low diastereoselectivity of products”? check it.

Author Response

Reviewer 4:

This review summaries asymmetric reactions of N-phosphonyl/phosphoryl imines developed by Li, Hoveyda and few other groups in the past decade, including the chiral auxiliary based induction strategy and asymmetric catalysis, to effectively access to enantio and diastereomeric amines. Given importance of chiral amines in synthesis of medicinal, pharmaceutical, and natural products, I believe this review will appeal to a wide range of readers and recommend a publication in Molecules. However, there are many grammar and format errors to be corrected before publication although the manuscript appears to be well-organized. For examples but not limited to,

1) The use of the singular and plural appears to be rather casual thorough out the manuscript. Too many to be listed here.

Response: The mistakes in singular and plural form has been cross checked and rectified as far as possible.

2) “The challenges associated with deprotection of these imines……”? should be “…… deprotection of these amines.”

Response: The change in the text from imines to amines has been done as per suggestion.

3) “significant yield of……” is not common.

Response: As “significant yield of……” is not acceptable, it has been changed to “high yield/ very good yield of…..” throughout the manuscript.

4) Many grammar errors like “Diverse N-phosphonyl auxiliary were examined…..” etc.

Response: The grammatical errors have been corrected as per reviewer’s suggestion.

5) Please capitalize the first letter in “grignard reagent” thorough out the manuscript.

Response: The “grignard reagent” has been corrected to “Grignard reagent” throughout the manuscript as per reviewer’s suggestion.

6) Please check the capitalization thorough out the manuscript, e.g. “concept through Asymmetric synthesis”, “application of Group-assisted purification” …… etc.

Response: The capitalization of the letters has been done accordingly.

7) “……stereoselectivity, hence owing to the low diastereoselectivity of products”? check it.

Response: The sentence is logically correct, however, it has been reframed as per suggestion to “selectivity. Hence, a low diastereoselectivity of products was observed in this case”.